# Sensory Involvement in Amyotrophic Lateral Sclerosis

**DOI:** 10.3390/ijms232415521

**Published:** 2022-12-08

**Authors:** Miguel A. Rubio, Mireia Herrando-Grabulosa, Xavier Navarro

**Affiliations:** 1Neuromuscular Unit, Department of Neurology, Hospital del Mar, 08003 Barcelona, Spain; 2Department of Cell Biology, Physiology and Immunology, Institute of Neurosciences and CIBERNED, Universitat Autònoma de Barcelona, 08193 Bellaterra, Spain

**Keywords:** ALS, small fiber, sensory, autonomic, proprioceptive, somatosensory

## Abstract

Although amyotrophic lateral sclerosis (ALS) is pre-eminently a motor disease, the existence of non-motor manifestations, including sensory involvement, has been described in the last few years. Although from a clinical perspective, sensory symptoms are overshadowed by their motor manifestations, this does not mean that their pathological significance is not relevant. In this review, we have made an extensive description of the involvement of sensory and autonomic systems described to date in ALS, from clinical, neurophysiological, neuroimaging, neuropathological, functional, and molecular perspectives.

## 1. Introduction

Amyotrophic lateral sclerosis (ALS) is a neurodegenerative disorder that involves both upper and lower motor neurons. The signs and symptoms derived from the involvement of the motor pathways are the ones that move the patient to consult for the first time, lead to functional decline, and, finally, to death, with a median survival of 3–5 years. Based on the clinical picture, it is not surprising that ALS has been classically considered an exclusively motor pathology. However, in recent years there has been growing evidence of non-motor symptoms, including cognitive, extrapyramidal, and sensory impairment [1,2].

Hereafter, we review the evidence of the involvement of the different sensory pathways in ALS, based on animal models, a series of clinical cases, neurophysiological studies, neuroimaging tests, and neuropathological studies. We include the somatosensory, visual, olfactory, gustatory, and auditory systems, as well as the autonomic nervous system.

## 2. ALS as a Purely Motor Disease

The traditional concept of a purely motor disease originates with the first descriptions of the disease. Despite not being the first to describe a motor neuron disease, Jean-Martin Charcot (Paris, 1825–1893) [3] was the first to use the term “amyotrophic lateral sclerosis” and to provide a very accurate description. Already in his first publication on the disease, he clearly described the absence of sensory symptoms in arms and legs, as well as the absence of apparent pathological alterations in the posterior columns of the spinal cord and the sensory peripheral nervous trunks.

This concept is also evident in the different diagnostic criteria used to date [4,5,6]. According to these criteria, sensory neurography should be normal (in both amplitude and velocity) unless an entrapment neuropathy or other known etiology could explain its abnormalities. The coexistence of a marked unexplained sensory neuropathy is therefore considered a red flag in ALS diagnosis. Beyond this assumption, the severe motor symptoms can easily overshadow other mild neurological manifestations, such as sensory and autonomic disturbances.

## 3. Non-Motor and Non-Sensory Alterations in ALS

Despite the markedly severe involvement of the motor nervous system, with loss of muscle strength as the main symptom, abnormalities have been also reported in extra-motor functions, defined as those that are not due to dysfunction of the upper or lower motoneurons or the corticospinal tract. Although it is unclear whether these extra-motor abnormalities occur in all patients, ALS may be considered a multisystem disorder, with a core set of motor symptoms required for diagnosis [7,8]. It is also unclear whether extra-motor abnormalities are associated with a specific pathology or specific causal mutations.

The spectrum of extra-motor alterations is wide and reinforces the idea that the disease is far from motor-centric [1]. Already in the earliest observations of Charcot [3], it was noted that there were special features in the skin of ALS patients; almost one century later, these were histologically confirmed and better characterized [9,10,11]. ALS patients in the later stages of the disease are usually immobilized and confined to a bed or a wheelchair, but they rarely develop bedsores, even in terminal stages [12]. This particular cutaneous feature can also be tested clinically by stretching the skin of the patient and observing what is known as a ‘delayed relaxation phase’. Skin collagen alterations are thought to be responsible for this feature. Collagen bundles in the dermis, especially in the papillary layer, have a small diameter, and are disoriented, fragmented, and separated by hyaluronic acid accumulations, as well as being less dense [13,14]. These observations are more marked as the disease advances. Moreover, there is an altered expression of several proteins (angiogenin, cystatin C, insulin growth factor 1, galectin 1, hepatocyte growth factor, progranulin, interleukin-6, laminin 1, TNF-alpha, ubiquitin, valosin-containing protein, TDP-43, and vascular endothelial growth factor) in the skin of ALS patients [15].

However, the most clinically relevant non-motor feature is cognitive impairment, present in approximately 30–50% of patients. Furthermore, 5–15% of ALS patients meet the diagnostic criteria of frontotemporal lobar degeneration (FTLD), predominantly the behavioral variant (bvFTLD) [16,17,18,19,20,21]. This comorbidity has been consistently linked to a worse prognosis [22,23]. The cognitive profile of these patients includes deficits in executive functions (especially phonemic verbal fluency, which occurs early), language, social cognition, and verbal memory [16]. Language production impairment may be disregarded due to confounding factors such as dysarthria and respiratory insufficiency. Some characteristic features are syntactic processing deficits, and semantic, verbal, and grammatical errors [24,25,26,27,28,29]. Behavioral changes are also described, with apathy being the most common feature (40–80%). Depression around the time of diagnosis is also common, and other traits can appear such as disinhibition, irritability, loss of empathy, and emotional lability [30,31,32].

Frontotemporal lobar degeneration explains cognitive impairment only in some ALS patients, while other features such as the extent of TDP-43 pathology and the presence of hippocampal sclerosis may also contribute to cognitive disturbances [33]. Neuroimaging studies have shown the involvement of extra-motor brain regions such as the hypothalamus [34], frontotemporal areas, cerebellum, and basal ganglia [35,36,37,38].

In addition, extrapyramidal features have been described in ALS patients. Almost 70% of patients present rigidity in the lower limbs [39]. This association was recognized in the original description of the ALS–parkinsonism–dementia complex (also known as Lytico–Bodig disease) in indigenous residents of Guam island [40]. ALS patients harboring pathological hexanucleotide expansion in the *C9ORF72* gene can also present atypical parkinsonian features, Huntington-like symptoms, and hemiballism [41,42,43]. Extrapyramidal symptoms have also been described in combination with ALS features in patients with the mutation A382T in the *TARDBP* gene [44]. Another interesting link between ALS and extrapyramidal symptoms is in the role of *ATXN2*. *ATXN2* pathological expansion of trinucleotides is linked to familial spinocerebellar ataxia type 2 (SCA2), but it is also a risk factor for ALS [45,46], and it is linked to TDP-43 proteinopathy [47,48,49]. Other non-motor symptoms, such as sleep disorders [50,51] and vestibular deficits [52], have been described as well.

## 4. Involvement of the Somatosensory System in ALS

Alterations in the somatosensory system are not unusual in ALS patients, although as noted above, abnormalities in sensory nerve conduction are regarded as an uncertain condition for the diagnosis. Thus, the recognition and characterization of sensory signs and symptoms are of clear importance in the knowledge of the etiopathogenic and clinical features of ALS. There are common links between sensory afferents and the motor aspects of the disease, since fasciculations, a characteristic and main feature in ALS, can be modulated by sensory nerve stimulation [53]. This peculiarity reinforces the fact that sensory pathways play a role in the clinical manifestations of the disease, and therefore should not be considered an isolated compartment, aside from the pathophysiological mechanisms occurring in motor pathways (Figure 1).

## 5. Somatosensory Cortex

Parallel to the loss of pyramidal neurons in layer V of the motor cortex, alterations of the somatosensory cortex have been described, mostly based on neuroimaging and pathological studies. Interestingly, in a cohort of 45 C9ORF72-negative ALS patients, 49% presented agraphestesia and 29% astereognosis [54]. MRI studies have shown a thinning not only of the primary motor cortex but also of non-motor areas such as the frontotemporal and parietal cortices [55]. This finding has been seen in *C9ORF72* and non-*C9ORF72* ALS patients [54] and ALS patients both with and without dementia, although cortical thinning is more pronounced in the presence of cognitive impairment [56,57]. As expected, sensory cortical involvement is more severe in classical ALS patients than in lower motor neuron-predominant patients [58]. Interestingly, decreased grey matter density of the parietal lobe (and other extra-motor regions) is associated with an increased central motor conduction time, a neurophysiological parameter that indicates a worse motor outcome [59].

Morphometric studies using fractal dimension (which enables the quantification of brain grey and white matter shape complexity) showed a widespread degeneration involving the primary sensory cortex, and, more importantly, these changes were more pronounced in those ALS patients with concomitant FTLD [60]. It has been proposed that there may be disturbances in the central plasticity of sensory–motor networks in ALS, based on the coexistence of ALS and complex regional pain syndrome in some patients [61]. Non-invasive in vivo indicators of neuronal loss, such as the reduction in N-acetylaspartate (NAA) and NAA:creatine assessed through MRI spectroscopy, are present in the primary sensory cortex, thalamus, basal ganglia, and other frontal and parietal regions [62,63]. Reactive gliosis is a common pathological feature accompanying neuronal loss in ALS. The use of certain radiotracers (11C(R)K119574 and 18F-DPA-71475) allowed the detection of increased activated microglia in the sensory cortex of ALS patients, as well as in motor cortex and midbrain [64,65].

Studies with functional MRI (fMRI) have also shown alterations in the form of increased functional connectivity in the somatosensory, prefrontal, and thalamus regions [66,67]. This increase in connectivity, which is also more pronounced in patients with primary lateral sclerosis (PLS), is attributed to a compensatory effect in the event of a deterioration in the functioning of these pathways. Despite this increased connectivity [18], FDG-PET studies have shown hypometabolism in the sensorimotor cortex correlated with the severity of motor symptoms in spinal-onset ALS patients [68].

Neurophysiological studies have shown interesting similarities between motor cortical and somatosensory projections. Cortical motor hyperexcitability is a well-known feature of ALS that is, furthermore, associated with survival [69,70]. Using neurophysiological techniques, there has also been found to be increased excitability of the somatosensory cortex in ALS. Shimizu et al. [71] reported an increased amplitude in cortical components of somatosensory-evoked potentials (SEP) evoked by stimulation of the median nerve (N25p-P25p; activity derived from the thalamocortical fibers to the primary somatosensory area, and subsequent projections through the thalamus to motor and premotor areas) that also correlated with a worse prognosis. Also, an increased sensory cortical excitatory activity has been suggested by the finding of loss of activity of cortical inhibitory interneurons, assessed with high-frequency somatosensory potentials [72]. In addition, an initially increased SEP amplitude (enlarged N20 response) has been described in some patients, followed by a progressive reduction until its total loss, as the disease advanced to a locked-in state [73]. In line with such dynamic changes, other investigators found a decrease in sensory cortical excitability (reduction in early and late SEP cortical components: N20, P25, N30, N60, and P100), but in this study, the peripheral component of SEP (N9) was also diminished [74].

Pathological studies have described neuronal loss in the somatosensory cortex [75,76], raising the question of whether alterations of the sensory cortex occur as a consequence or an extension of motor cortex impairment, or rather whether there may be specific and independent pathologic changes in the sensory cortex. In this regard, clues from TDP-43 proteinopathy indicate that there is a sequential propagation of pTDP-43 accumulation, disseminating from the agranular neocortex and bulbar motor neurons, and finally reaching sensory neocortical areas of parietal, temporal, and occipital lobes in advanced stages (stage 3 of TDP-43 staging system) with a corticofugal spreading pattern [77].

## 6. Spinal Ascending Sensory Pathways

Parallel to the pallor of the corticospinal tract in the spinal cord that gives its name to the disease, sensory ascending tracts that integrate different sensory modalities are also affected in ALS. In fact, combining neuroimaging and neurophysiological data, subclinical sensory alterations of spinal sensory ascendant pathways are found in up to 85% of patients [78]. Aside from the abovementioned application in the study of the cortical component in sensory integration, SEPs have been used mostly to assess the integrity of the dorsal column pathways of the spinal cord in ALS, by stimulating both the tibial and median nerves [79,80,81,82]. Alterations in amplitude and/or latency were consistently found in 33% to 69% of patients, more in SEPs elicited by the stimulation of lower extremity nerves. However, most patients with altered SEPs remained asymptomatic, suggesting that this impairment does not have enough magnitude to cause clinical manifestations.

Focusing specifically on dorsal columns, diffusion tensor imaging (DTI) showed the same alterations in cervical proprioceptive pathways as in the lateral corticospinal tract, in both sporadic and *SOD1* familial cases. No volumetric data of these regions are available to date, due to technical issues owing to the size of the region of interest. DTI analysis shows a decrease in the anisotropy fraction and increases in radial diffusion, and a decrease (although not significant) in the magnetization transfer sequences (reflecting a lower myelin content) of dorsal columns of the cervical segment [83]. DTI metrics of ascending sensory pathways correlate with SEP amplitude alterations in the N9 component [78].

These non-invasive and in vivo studies confirm that these alterations are more marked with disease progression, suggesting that its involvement advances in parallel to motor neuron degeneration. Regarding its clinical impact, it is estimated that between 18% and 50% of ALS patients may present some alterations of vibratory sensitivity on physical examination that is not attributable to another cause [84,85,86], and in rare cases leading to clinical sensory ataxia.

Pathological studies confirmed that the degeneration of the dorsal columns occurs in up to half of patients, especially in lumbar regions, including both genetic (*SOD1*) and sporadic forms [86,87,88,89,90,91,92,93,94,95,96,97,98,99,100]. The predominant involvement of lumbar spinal cord segments in both functional and pathological examinations suggests an axonal length-dependent degeneration, taking as a starting point cortical and subcortical brain structures, as suggested by the corticofugal propagation proposed by Braak [77]. Supporting this hypothesis of cortical influence in the degeneration of spinal sensory ascending axons is the fact that dorsal column degeneration is less frequent (8%) in pure progressive muscular atrophy (PMA) and in those patients that progress to ALS (30%).

## 7. Dorsal Root Ganglia and Dorsal Roots

Dorsal roots and dorsal root ganglia (DRG) have been studied mostly in animal models of the disease. However, descriptions of necropsies have also reported an axonal loss in the dorsal root and DRG, especially of large nerve fibers [91]. In the familial form of ALS secondary to the p.A382P mutation of the *TARBPD* gene, a clinical sensory neuronopathy was confirmed by histopathology [92]. Interestingly, a case has been reported of an ALS patient with sensory neuropathy/neuronopathy and with biallelic RFC1 AAGGG repeat expansion (400) [93]. RFC1 biallelic expansion is related to CANVAS, a rare genetic disorder characterized by cerebellar ataxia, sensory neuropathy/neuronopathy, and vestibular areflexia. This raises the question of whether motor neuron disease is part of the spectrum of that particular form of cerebellar ataxia and sensory neuronopathy, as is the case with *ATXN2* spinocerebellar ataxia type 2 (SCA2).

Studies related to the sensory pathways of SOD1 transgenic mice, the most commonly used murine model of ALS, have demonstrated pathological changes in the axons of the dorsal columns, the dorsal horn of the spinal cord, and dorsal root, and in the soma of DRG neurons [94,95], similar to what is observed in Wallerian degeneration. In addition, an anomalous accumulation of SOD1 protein in sensory axons and the soma of DRG neurons has been reported [96]. These alterations were mostly seen in large axons and were already evident at an early stage in mice (60 days of life) when they are still asymptomatic from the motor point of view.

In vitro studies of DRG neurons of SOD1^G93A^ mice showed the accumulation of peripherin, a splice variant of peripherin that disassembles light and medium neurofilaments, resembling axonal stress features that appear in motor neurons [97]. In cultured sensory neurons harboring known ALS-causing mutations, SOD1^G93A^, and TDP43^A315T^, neurite growth was slower and showed increased sensitivity to cell stressors like vincristine. Although both neuronal cultures showed abnormalities, neurites grew significantly slower in TDP43^A315T^ neurons and were more sensitive to vincristine than SOD1^G93A^ cells [98]. However, although subcellular and functional changes have been described, sensory DRG populations are preserved over time [99] while their distal axons in the skin are already reduced [100], suggesting distal sensory axonopathy, similar to what has already been described in the motor component of the disease [101] (Figure 2).

In addition to the sensory features of the SOD1^G93A^ mouse, a non-transgenic animal model of motor neuron disease, a neurodegenerative disease called canine degenerative myelopathy, also produces sensory symptoms in addition to motor impairment. This disorder, which occurs in several adult dog breeds, is characterized by the development of proprioceptive ataxia and spastic paraparesis of the hind legs, which over time progresses to flaccid tetraplegia and the appearance of dysphagia. In 12 months, the animal typically becomes quadriplegic, although this time can vary from 6 to 36 months. Pathological studies showed the degeneration of motor neurons at the anterior horn as in ALS disease, but also the loss of somas at the DRG and axons in the sensory roots. Interestingly a missense mutation in the *SOD1* gene has been described as responsible for the disease [102,103].

## 8. Peripheral Nerve Trunk

Although previous diagnostic criteria only admitted very mild alterations in sensory nerve conduction studies, an abundance of data indicates the impairment of sensory peripheral nerves in ALS. Up to 32% of patients may have subjective sensory symptoms in the form of a glove and sock hypoesthesia, paresthesia, or dysesthesia, resembling the typical manifestations of length-dependent axonal neuropathies [86]. The results of a multicenter study showed that 12.5% of patients met electrophysiological criteria for sensory polyneuropathy of unknown origin [104]. In addition, up to 10.3% of patients may have alterations in the sensory conduction studies of some nerves without reaching the criteria of polyneuropathy, and without a proven etiology. Other studies estimate that neurophysiological sensory alterations are present in between 13% and 22% [105,106,107,108], and relatively more frequent (38%) in *C9ORF72*-ALS patients [109]. Such alterations in sensory neurography can affect the amplitude and/or conduction velocity. As an example, the sural nerve, which is the most studied nerve, shows a reduction in conduction velocity in 14–27% of cases [106,110], a reduction in amplitude in 6–27% [86], and a reduction in both amplitude and velocity in 17% [111]. The longest axons seem more commonly to present such abnormalities since the exploration of distal sensory nerves (dorsal sural and medial plantar nerves) has shown alterations in nerve conduction studies in up to 66.7% of ALS cases [112]. With a different approach, based on the use of near-nerve electrodes, slow sensory conduction velocity has been reported in 50% of ALS patients [107].

A retrospective study that reviewed electrodiagnostic data from 99 *C9ORF72*-ALS patients found that they have higher median sensory nerve potentials compared to non-C9-ALS patients, although the authors concluded that these differences were not clinically relevant enough to differentiate this genetic subgroup in a clinical setting [113]. Sensory neuropathy has been described in both C9 and non-C9-ALS subjects, and while some have described these abnormalities as more frequent in males regardless of the genetic background [113], in the case of *C9ORF72*-ALS patients, sensory involvement is present most frequently in male (80%) and those with spinal onset (67%) [109].

SOD1^G93A^ mice also have a slowing of sensory conduction velocity [114]. In transgenic ALS mice, sensory myelinated axons present normal membrane properties and excitability within normal values, as assessed by threshold tracking techniques, in contrast to motor axons [115].

Although in most cases these alterations are mild or even subclinical, in certain genetic forms sensory peripheral involvement is characteristic and clinically relevant. Familial ALS cases associated with the Gly93Ser mutation of the *SOD1* gene may suffer urinary alterations and sensory neuropathy [116]. In the FOSMN syndrome (facial-onset sensory motor neuropathy) patients present trigeminal neuropathy, and facial and bulbar involvement. Although FOSMN is considered a motor neuron disease distinct from ALS, heterozygous mutations of the *SOD1* gene (D90A mutation) suggest a link to classical ALS [117].

Pathological studies have confirmed that sensory nerves, such as sural and superficial peroneal nerves, present axonal alterations in ALS, with up to 91% of the sural nerves having a significant loss of axons [86,118,119,120,121]. This loss involves predominantly large fibers (73%), although small myelinated axons (23%) and unmyelinated fibers are also affected. Moreover, there is evidence of a reduction in axonal transport in sensory axons of sural nerves [110]. Similar results have been obtained in sural nerves of pseudopolyneuritic forms of ALS (Patrikios’ syndrome), showing also Wallerian-like degeneration, and an increased number of mitochondria and neurofilaments [122,123].

Some demyelinating changes were shown in teased fibers, but clearly to a lesser degree when compared to axonal degeneration. The myelin lesion was suggested to be a consequence of alterations in the interaction between Schwann cells and axons [85,124] and a result of the dynamic physiological changes in axonal injury and repair [125]. A detailed pathological study of sural biopsies from ALS patients showed an increase in clustered remyelinated internodes in parallel with disease duration [126]. In the same study, although an increased number of myelin lamellae was found, the diameter of the nerve fibers (axon and myelin sheath) was significantly smaller than in control nerves. Disturbances of the large and therefore fastest axons could explain slow conduction velocities in sensory neurography, rather than a primary myelin or membrane problem. Additionally, the major involvement of the longest nerves, as suggested by both pathological and neurophysiological studies, suggests axon transport deficiencies [110] and agrees with the “dying-back” process of ALS [127].

In mixed nerves, cutaneous sensory and muscular motor fascicles are in proximity, separated only by the perineurium, but even muscular fascicles contain motor and sensory axons innervating the skeletal muscle. ALS patients and animal models present a low expression of occludin [128,129], a tight-junction protein present in the perineurium that contributes to the blood–nerve barrier maintenance, as well as in the blood–spinal cord barrier in the CNS [130]. This suggests that the blood–nerve barrier could be compromised and therefore allow leakage between motor and sensory fascicles in peripheral nerves. Nevertheless, whereas in motor peripheral axons (and their Schwann cells), deposits of cytoplasmic phosphorylated TDP-43 have been described in a large cohort of predominantly lower motor neuron ALS [131]; this feature has not been explored or reported in sensory axons.

## 9. Proprioceptive Afferences

Up to 25% of ALS patients have clinically relevant proprioceptive deficits [132] with postural abnormalities and problems with axial control [133]. A study combining posturography and neurophysiological examinations found that stance control in ALS patients was not affected by paresis or reflex hyperexcitability [134]. Although manifestations of this nature could be attributed, at least in part, to the aforementioned involvement of neurons in the DRG, there are data supporting the involvement of proprioceptive afferents. However, up to 68% of patients present a diminished laryngeal adduction reflex caused by a loss of small sensory fibers of the epiglottis [135].

The degeneration of sensory fibers Ia/II of the muscle spindles has been observed in SOD1^G93A^ and TDP43^A315T^ mutated mice at a presymptomatic stage [136] but with the preservation of the neuron’s soma in the DRG. Moreover, the sensory denervation of intrafusal muscles was more marked as the disease progressed. In the case of the TDP43^A315T^ mouse, this finding was independent of alpha-motor neuron degeneration, while in the SOD1 mice these changes were observed alongside motor neuron loss. Overall, these findings suggest that the degeneration of proprioceptive endings occurs independently of the defects in the motor system. Interestingly, the authors reported muscle spindles innervated by gamma-motor neurons without Ia/II fibers, near alpha-motor endings retracting from neuromuscular junctions in extrafusal muscle fibers. At the symptomatic phase of the SOD1^G93A^ mouse, proprioceptive synapses in the anterior horn of the spinal cord were reduced, but not in the TDP43^A315T^ mouse. This loss of central synapses appeared later than the alterations in the peripheral endings at muscle spindles. In contrast, Ib fibers (the afferents from the Golgi tendon organs) were mostly preserved (Figure 2).

Moreover, a detailed electrophysiological study analyzed Ia afferents of jaw proprioceptive sensory neurons (jaw reflex) of a SOD1^G93A^ mouse and showed electrical abnormalities with an impaired excitability of these fibers and a reduction in the voltage-gated Na+ currents (Nav1.6 channel) reflecting the vulnerability of the proprioceptive–reflex circuit [137]. Similar evidence has been found in the Drosophila SOD1^G85R^ model [138], with an impairment of the sensory feedback even before motor neuron degeneration is established.

Other neurons that participate in the spinal circuitry and proprioceptive integration are also known to degenerate in ALS; that is the case of Renshaw cells [139,140], commissural neurons (important for bilateral coordination and locomotion) [141,142], intermediate neurons of Clarke’s column (lamina VII), neurons of the lateral cuneate nucleus, and spinal border cells [143,144,145,146].

The close interaction between proprioceptive afferences and alpha-motor neurons, in addition to the loss of interneurons between them, suggests that neurodegenerative mechanisms could reciprocally influence each other. In this regard, SOD1 mice that also carry dominant mutations in cytoplasmic dynein (a protective factor for motor neuron loss) significantly present a slow degeneration with a practical absence of alpha-motor neuron loss at 18 months and with a marked loss of proprioceptive sensory axons [147]. This finding opens the question of whether the proprioceptive fibers could play a role in neurodegeneration, mediated by glutamatergic excitatory inputs to motor neurons.

In line with this, gamma-motor neurons have proven to be resistant in ALS, and the lack of synaptic contact from primary Ia afferent fibers is one interesting distinction with the most vulnerable alpha-motor neurons. In addition, the functional reduction in Ia activation delays symptom onset and prolongs the lifespan of the SOD1^G93A^ mouse [148].

In other motor neuron disorders, such as in the spinal muscular atrophy (SMA)-mouse, there is a reduction, also at early stages, of the synapses between primary afferent proprioceptive neurons and spinal motor neurons [149]. This loss of proprioceptive synapses is more severe in motor neurons projecting to proximal muscles, the ones most affected by the disease and is the earliest and most pronounced pathological feature of SMA-mice. Although it is hypothesized that the loss of these synapses is due to a failure in postnatal maturation, similarities with ALS point to other shared mechanisms involved in the impairment of sensory–motor connectivity.

## 10. Small Sensory Nerve Fibers

In humans, although there are no specific studies on the involvement of small nerve fibers, published data show that 50–78% of ALS patients experience pain [150,151,152,153], and 20–30% described it as “electric” or “burning”, which may correspond to symptoms secondary to nociceptive fiber involvement. However, although these characteristics resemble small fiber neuropathy symptoms, those sensations have been described mostly in the lumbar and proximal regions of upper limbs, in sharp contrast to the distal limb pattern expected in classical length-dependent small fiber neuropathies.

Studies based on neurophysiological techniques have shown controversial results. While no changes have been detected using contact heat-evoked potentials (CHEPS) [154], amplitude alterations were obtained using laser-evoked potentials [155]. Thermal quantitative sensory tests (QST) (another technique for the study of pain thresholds and thermal sensations of cold and heat) have also shown contradictory results [156,157], in some cases showing alterations in the threshold of the heating stimulus, especially in spinal forms [158]. Nevertheless, the differences in results between studies suggest technical issues. It must be pointed out that nociceptive-evoked potentials are based on the stimulation of Aδ type II fibers [159] and not of C-fibers, while thermal studies are based on the assessment of both Aδ and C axons.

More consistent are the results of the study of unmyelinated fibers in skin biopsies (Figure 2). ALS patients have a lower density of intraepidermal nerve fibers (IENF), with up to 79% of patients meeting the criteria for small fiber neuropathy [160,161]. These alterations are not only in number, but also in morphology, with axonal swellings suggesting an alteration in axonal transport, and insufficient growth of terminal branches [162]. This finding is not only seen in sporadic ALS patients, but also genetic forms including *SOD1*, *C9ORF72,* and *SQSTM1*.

Some authors have found differences between bulbar and spinal-onset forms, with lower IENF density in the latter [158], but other groups have not found differences between onset type, phenotype, clinical course, and severity [163]. TDP-43 deposition or its phosphorylated form has not been found in IENF [164]; it is found only in autonomic dermal fibers [165]. In the SOD1^G93A^ mouse, a loss of small fibers has been seen in the skin, not only limited to the epidermis [97,166] but also at the level of sensory axons in the dermis and involving Meissner corpuscles, with evidence of retrograde sensory axonal degeneration [167]. In this animal model IENF impairment occurs early, before motor symptoms (Figure 3), and while both peptidergic and non-peptidergic axons are impaired, initially non-peptidergic fibers are the most affected [100].

The loss of small fibers is not restricted to the skin. The use of corneal confocal microscopy has allowed the study of small fibers in vivo, with a non-invasive, repeatable, and quantitative analysis of small fibers from the trigeminal nerve. ALS patients have a decreased number of corneal nerve fibers and morphological alterations with increased tortuosity and decreased size. The degree of bulbar involvement (quantified according to the ALSFR-R bulbar subscale) was inversely related to the length of these fibers [168]. However, similar to what was found in skin biopsies, there was no relationship with the duration of the disease, the age of the patient, or the degree of impairment of upper or lower motor neurons.

Interestingly, a diminished sensitivity of the larynx has also been reported in up to 54% of ALS patients [169]. One study in 114 patients found laryngeal sensory deficits when exploring the laryngeal adduction reflex (with preserved functionality of the adductor muscles), with impairment in bulbar-onset patients being more frequent. Three of these patients underwent a biopsy of epiglottis mucosa; in one there was a total absence of intraepithelial innervation and in the other two there were pathological changes in the remaining fibers, including axonal swellings, chaotic nerve branching, and branches not crossing the entire epithelium [135]. In another study, ALS patients with confirmed silent aspirations presented impaired sensory responses when exploring the tussigenic reflex, suggesting the degeneration of the airway sensory receptors [170].

While many of these findings could be explained by a longitudinal dependent pattern and a distal sensory axonopathy, it is worth noting that small fiber impairment is not always limited to the most distal regions, and in some cases, a similar loss of IENF has been found between proximal and distal regions of the limbs [161]. Nevertheless, the existence of a sensory ganglionopathy could explain such a pattern [171].

## 11. Involvement of the Visual System in ALS

Although some studies have shown the presence of abnormal eye movements [172,173,174,175], the oculomotor function is classically considered to be preserved in ALS. In contrast, there is abundant evidence of the involvement of the visual sensory pathway. A substantial part of this evidence comes from histopathological studies or imaging tests since the results of functional studies, as expected from clinical observations, are minimal or non-existent, and often yield contradictory conclusions [176]. Visual field studies seem to be unreliable due to motor difficulties inherent to the disease (fixation losses, false positives, and negatives), and only one study reported impairment of the visual field in a cohort of ALS patients [177]. Visual acuity (high and low contrast) is commonly preserved [178,179,180], although one study showed low visual acuity in both high and low contrast using Sloan charts [181], and another described an impaired contrast sensitivity in some *C9ORF72* patients [182]. Visual evoked potentials have shown normal values of amplitude and latency [183,184], but mildly increased latency or other minimal abnormalities have also been reported [185]. fMRI neuroimaging showed a reduction in the response to visual stimulation in secondary visual areas [186].

The histopathological examination of the visual pathway has revealed a remarkable difference between the involvement of retrochiasmatic structures and the anterior segment of the visual pathway. The tractus opticus, corpus geniculatum lateralis, colliculus superior, radiation opticus, and the primary visual cortex are preserved in advanced ALS patients [76] and SOD1^G93A^ mice [187], although reactive astrogliosis has been shown in occipital areas [188]. However, recent morphometric neuroimaging studies have shown that some components of the visual pathway may be affected: the volume and thickness of the lateral occipital cortex (involved in facial and object recognition), the volume of the right lateral geniculate nucleus, and the lateral posterior nucleus [54].

In contrast, ganglion cells in the retina seem to be the most vulnerable neuron type of the visual system in ALS. The assessment of the thickness of different retinal layers, including the ganglion cell layer (GCL) and its long axons, and the retinal nerve fiber layer (RNFL), can be easily done with optical coherence tomography (OCT). OCT is a non-invasive and objective technique that provides a high-resolution image of the retina, and it has been shown to be useful for the assessment and monitoring of optic nerve damage in various neuroinflammatory and neurodegenerative diseases, such as Parkinson’s disease, Alzheimer’s disease, and multiple sclerosis [189,190]. Using OCT, studies have consistently reported thinning of the RNFL in ALS compared to controls [191,192,193,194]. Only a few studies have not found differences in the OCT measurements. These contradictory results could be explained by methodological aspects, such as the heterogeneity in the technology used [195], but also by the different profiles of ALS patients [196] and control selection [177]. Additionally, early retinal thinning also occurs in individuals with FTLD due to progranulin (*GRN*) mutations, a gene that is involved in dominant forms of ALS and ALS-FTLD [197].

Interestingly, one study also showed an increased thickness in certain areas of the retina [180], suggesting alterations in axonal transport, protein aggregation, and neuroinflammation as possible explanations. Supporting this finding, in post-mortem studies, a significantly greater number of PAS-positive spheroids has been found in RNFL, especially in the peripapillary region [198]. The increased presence of phosphorylated neurofilaments in these spheroids could be explained by a special vulnerability of the long axons to axonal transport deficits. In addition to the apparent vulnerability of ganglion cells in the disease, a critical role of Ranbp2 in the signaling between microglia and retinal ganglion cells in the immune response in ALS has been elucidated [199]. Ranbp2 is a protein involved in nucleo-cytoplasmic transport whose regulation is impaired in both sporadic and familial ALS [200].

Other retinal cells, rather than ganglion cells, and layers have been found to be impaired, although findings are less constant than in the RNFL. In an animal model of an X-linked form of ALS (*UBQLN2*), ubiquitin2 positive aggregates are found predominantly in the inner nuclear layer (INL), but a few are also found in the outer plexiform layer (OPL) and the GCL [178]. In the SOD1^G93A^ mouse model, vacuolization occurs in the excitatory dendrites of retinal neurons of the inner plexiform layer, as well as in the GCL and INL [187].

Findings in other genetic forms of ALS support the involvement of other retinal cell subtypes, suggesting that those impairments could be gene-specific. P62-positive and pTDP43-negative perinuclear inclusions have been described in *C9ORF72*-ALS patients in bipolar cells (located in the INL), GCL, amacrine, and horizontal cells [182], with the colocalization of poly-GA dipeptide and ubiquitin in such inclusions. Moreover, *senataxin* (*SETX*) ALS could involve, more profoundly, photoreceptors. *SETX* mutations are involved both in recessive forms of ataxia-oculomotor apraxia 2 (AOA2) and in a dominant juvenile onset form of ALS (ALS4). In the ALS4 phenotype, it is hypothesized that neurodegeneration occurs as a result of a partial gain-of-function of the senataxin protein, whose functions are related to nucleic acid processing. Although senataxin is ubiquitously expressed in many tissues (including lens and retina), its levels are cell-specific and it is highly expressed in inner and outer segments of photoreceptors and outer plexiform layers, but poorly presented in inner plexiform and ganglion cell layers [201].

Those findings in the retina raise the question of whether they are secondary to retrograde trans-synaptic degeneration or due to primary retinal damage. Consistent with the latter is the fact that pathological examination did not find substantial impairment in the rest of the visual pathway, in contrast to what is described in other cortical non-motor structures. Also, in favor of a primary retinal involvement, vascular changes have been found in the retina of ALS patients [202], as in other anatomical regions, suggesting shared mechanisms of degeneration.

In fact, there are several similarities in the pathways involved in ALS neurodegeneration and certain optic nerve disorders, such as increased levels of oxidative stress, axonal transport deficits, and mitochondrial damage. Several genes associated with familial forms of ALS (*optineurin*, *TBK1,* and *ataxin2*) are also associated with chronic primary open-angle glaucoma, suggesting common mechanisms, as in the case of *optineurin*, which is involved in neuroinflammation, vesicular trafficking, and autophagy.

## 12. Involvement of the Olfactory System in ALS

Olfactory dysfunction appears with normal aging [203,204] with a prevalence between 6% and 16% in healthy elderly people [205]. It is also a common feature in neurodegenerative disorders and constitutes part of the prodromal manifestations of Alzheimer’s and Parkinson’s disease [206,207]. Between 46% and 75% of ALS patients present a decreased ability to identify odors, and 11% have complete anosmia, based on psychophysical tests, mainly the University of Pennsylvania Smell Identification Test (UPSIT) [208,209,210,211,212,213]. This feature seems especially common in the subset of patients with behavioral and cognitive impairment [214]. Although olfactory deficits have been found in respiratory-spared ALS patients, it has been observed that the smell detection threshold is influenced by respiratory impairment [215], probably because of reduced inspiratory flow. Functional results differ from the results of olfactory-evoked potentials in which no pathological results were obtained, although one-third of patients could not be tested due to technical difficulties related to the disease [210]. Other authors using different functional tests did not find differences in smell identification between ALS and healthy controls [216].

Despite the similarity with other neurodegenerative diseases, histopathological studies have shown a very different pattern from that observed, for instance, in Parkinson’s disease. No loss of neurons has been found in the olfactory pathway [76] but a lower volume of the amygdala [217,218], and the presence of pTDP-43 inclusions in central structures of the olfactory pathways, including dentate granular cells, transentorhinal cortices, corticomedial and the basolateral part of the amygdala, anterior olfactory nucleus, and the piriform cortex have been found [211,219]. Quantitative analysis showed a gradient of p-TDP-43 inclusions, most frequent in the hippocampus and least in the olfactory bulb, with an intermediate density in the olfactory cortex. Based on this observation, it has been proposed that in the fourth and final stage of the TDP-43 neuropathological staging [220], cortical pathology progresses in the temporal lobe and reaches the allocortical entorhinal region as well as the hippocampal formation, and then spreads centrifugally from the hippocampus to the primary olfactory center, and later to the olfactory bulb [211], which also presents lipofuscin deposits [210]. This pattern differs from Parkinson’s disease, in which alpha-synuclein pathology involves mainly the olfactory bulb with centripetal spreading [221].

Interestingly, SOD1^G93A^ mice show vacuolization without inflammation in the excitatory dendrites of the granule cell layer of the olfactory bulb. This is considered a sign of neurodegeneration, present at the presymptomatic stage, which increases over time until the end-stage, but with the preservation of the anterior olfactory nucleus olfactory tract, and olfactory tubercle [187].

Olfactory neurons located in the olfactory bulbs of healthy humans express proteins involved in several neurodegenerative diseases, such as alpha-synuclein, beta-amyloid, and TDP-43 [222]. Moreover, a proteomic analysis of the olfactory bulb of ALS patients showed an aberrant expression of proteins involved in autophagy, axon development, and vesicle-transport [223]. In line with this, cells of the olfactory mucosa of ALS patients can induce disease-specific changes and decrease neuronal survival when co-cultured with human spinal cord neurons, and they also induce a glial inflammatory response [224].

The widespread impairment of olfactory pathways implies shared mechanisms with classical motor neuron degeneration beyond the TDP-43 pathology. For example, senataxin, a protein associated with a dominant juvenile form of ALS (ALS4) through a gain-of-function mechanism, has its greatest expression in the hippocampus and olfactory bulb [201]. Also, a transcription factor, Runx1, which is expressed in somatic motor neurons in the murine brainstem and cervical spinal cord, plays a significant role in the proliferation and development of olfactory ensheathing cells and the transition between undifferentiated neural progenitor cells and neurons in the olfactory epithelium [225].

## 13. Involvement of the Gustatory System in ALS

Taste perception problems are present in several central nervous system disorders, such as Alzheimer’s disease and mild cognitive impairment [226], Parkinson’s disease [227], stroke [228,229], and major depression [230], but are also found in 1% of the general elderly population [231]. In ALS, a reduction in taste may be present, more frequently in patients with an enteral tube [232], and in some cases appears restricted to bitter and sweet tastes [233]. However, some authors did not find gustatory impairment in ALS populations based on functional tests with taste strips [216]. In a small cohort of 45 C9OFR72-negative ALS patients, 24% self-reported changes in taste since the onset of the disease [54].

A limited number of histological studies focused on central gustatory pathways showing consistent proof of its involvement. The fact that patients on enteral nutrition have more hypogeusia/ageusia suggests that there is also a component of peripheral involvement, but to date, the scarce histological studies in this regard have not reported abnormalities [233]. However, severe neuronal loss has been described in the nucleus parabrachialis medialis (the region that carries gustatory information from the solitary nucleus to the ventral posteromedial nucleus of the thalamus) and the tractus trigeminothalamicus dorsalis [76].

Neuronal loss, reactive astrocytosis, and TDP-43 deposition can also be found in the motor nuclei of the trigeminal and facial nerves, the dorsal motor nucleus of the vagus nerve, and the nucleus of the hypoglossal nerve. TDP-43 pathology in the solitary nucleus is rare [234], and neuronal loss in this region has been reported in *FUS*-ALS patients [235].

TDP-43 pathology in gustatory neural pathways has also been explored in facial onset sensory and motor neuronopathy (FOSMN), another degenerative motor neuron disorder related to ALS, which can also present taste disorders [236]. In these cases, TDP-43 inclusions are present in motor and sensory nuclei of the trigeminal and facial nerves, the dorsal motor nucleus of the vagus nerve, the nucleus ambiguous, the nucleus of the solitary tract, and the hypoglossal nerve [237,238,239].

## 14. Involvement of the Auditory System in ALS

Information on the involvement of the auditory pathway in ALS is scarce, and no functional data are available. Neuroimaging studies have shown a volume and thickness reduction in the superior temporal and transverse temporal cortices, as well as in the right medial geniculate nucleus [54]. Histopathological examinations have shown a significant neural loss in brainstem regions involved in hearing, including the nucleus olivaris superior, the lemniscus lateralis, and the colliculus inferior.

In contrast, superior structures such as the corpus geniculatum medialis and gyrus temporalis transversus are relatively well preserved. Although this may point to a more peripheral involvement, the dorsal cochlear nucleus, the immediate relay structure of the auditory nerve in the brainstem, is preserved in ALS [76].

## 15. Involvement of the Autonomic Nervous System in ALS

Autonomic disturbances are rarely described in classical forms of ALS; however, observations from functional tests and self-reported questionnaires indicate that autonomic manifestations, although mostly mild, are not so uncommon [240,241]. Several studies investigating the subclinical occurrence of dysautonomia have reported dysfunction of cardiovascular, sudomotor, urinary, gastrointestinal, and salivary visceral regulation, even in early ALS cases, causing symptoms that may further deteriorate the state of health. Autonomic disturbances may even lead to arrhythmia, circulatory collapse, or sudden death in advanced-stage ALS patients [242,243].

The cardiovagal function was found to be impaired in up to 50% of ALS patients [241], more frequently in those with a low respiratory capacity [244,245]. Some studies have found an abnormally increased cardiovascular sympathetic tone in ALS patients [246] and SOD1^G93A^ mice [247,248], especially regarding alpha-sympathetic rather than beta-sympathetic hyperactivity. Additionally, there is increased activity of the norepinephrine transporter in cardiac sympathetic nerve terminals, which is indeed associated with shorter survival [249]. ALS patients present significantly increased plasma levels of norepinephrine, especially those that are bedridden [250]. Preganglionic sympathetic denervation may result in inappropriate levels of norepinephrine, but those increased levels can also be caused by impaired reuptake, secondary to nerve terminal degeneration. Moreover, as opposed to the resting sympathetic hyperactivity in ALS, there also exists a downregulation of alpha-adrenoceptors, providing a ‘ceiling effect’.

Nevertheless, this increased cardiovascular sympathetic drive is not observed in all patients at the same stage [251], and it seems to be more related to the flail arm/leg phenotype [252]. This is in line with the concept that muscle atrophy can lead to a diminished number of arterioles and vessels involved in vascular resistance, and therefore an insufficient capacity to accommodate temporary increases in blood flow [252]. In this regard, although rare, some patients can present hypertensive crisis, alternating with nocturnal hypotension without tachycardia, and occasionally leading to circulatory collapse and sudden cardiac arrest, especially in those who are respiratory-dependent [253]. Importantly, cardiovascular sympathetic activity decreases as disease advances [250,254,255], a deterioration thought to be due to the impairment of central sympathetic mechanisms rather than locally altered responses [256]. Nevertheless, vessel axons show degeneration and ultrastructural abnormalities (swelling and vesicle accumulations) [257], and sympathetic fibers present a decreased firing rate in the control of vascular resistance in ALS patients [258].

In addition, there is a cardiovascular parasympathetic hypoactivity with a decreased vagal tone [259,260,261,262,263], also during sleep [264]. Although autonomic cardiovascular responses are often impaired, orthostatic hypotension is rarely present in ALS [241]. The absence of symptomatic orthostatic hypotension is in contrast to other neurodegenerative diseases, such as Parkinson’s disease and multisystem atrophy, but this could be explained by the severe motor impairment in ALS that limits physical demand.

Microneurographic studies revealed abnormalities of the spontaneous and reflex activity of sympathetic efferents in ALS patients, suggestive of autonomic nervous system involvement and neurodegeneration [265]. The muscle sympathetic nerve activity (MSNA) at rest was found to be higher than in healthy subjects and patients with other neuromuscular disorders, and at early but not at advanced stages of ALS [266]. However, this elevated resting MSNA responded more weakly to activating maneuvers in ALS patients compared to controls [258]. In contrast, resting sudomotor and vasoconstrictive skin sympathetic neural activity (SSNA) was also found to be significantly greater in ALS patients than in healthy controls, but again with smaller responses to activating maneuvers such as mental arithmetic tests [266]. ALS patients also exhibited a slight prolongation of SSNA reflex latencies. All these observations suggest that sympathetic activity is abnormal in ALS patients, and mainly consists in resting hyperactivity that may be induced by central autonomic drive.

Sudomotor function has been studied, and up to 33% of ALS patients showed sweating abnormalities [165]. Quantitative studies of sudomotor regulation in different stages of the disease have shown that there is a greater activity with hyperhidrosis (especially in the palmar region) at early stages, but as the disease progresses, there is a significant decrease of about 40% in sweat production [250,267], and in 25% of cases with marked asymmetry [268]. A higher sweat rate near the onset of the disease could be explained by hypersensitivity to the partial denervation of sweat glands. In this regard, a higher sympathetic firing rate was found by microneurographic measurements in mildly disabled ALS patients, compared with equally disabled patients with other neuromuscular disorders [269,270].

In contrast, decreased sweat production could also be explained by the degeneration of postganglionic sympathetic fibers and atrophy of the sweat glands. Sympathetic skin response (SSR) testing has consistently confirmed this impairment by a diminished or even absent response, mainly in the lower limbs, and independently of possible influencing factors such as muscle wasting, inactivity, and skin temperature [258,271,272,273,274], although some authors did not find SSR impairment in the palmar region [275]. Similar impairment in sweat function has been described using a thermoregulatory sweat test and quantitative sudomotor axon reflex test (Q-SART) [276]. Interestingly, Q-SART showed that sweat impairment may be present in different forms, most frequently in a distal or length-dependent pattern (54%), but also as a patchy (32%) or more diffuse involvement (14%) [241].

A closer look at sweat glands shows that innervating axons present swellings, degeneration, TDP-43 accumulation [165], and synaptic vesicle accumulations, as well as presenting increased intracytoplasmic lipofuscin granules [257]. Pilomotor nerve fiber density is also reduced as there is profound denervation of arrector pili muscles in ALS. This finding is significantly more notable in PLS patients, raising the question of whether this is phenotype-dependent or merely indicates a longer disease duration [161]. In contrast to what is observed in the cardiovascular system, all of these observations suggest the involvement of unmyelinated postganglionic sympathetic fibers. Additionally, in the SOD1^G93A^ mouse, there is a loss of sudomotor fibers innervating sweat glands, although this occurs later than the involvement of IENF, already in the motor symptomatic stage [100].

Urinary autonomic disturbances have also been described, with a mild degree of neurogenic bladder in up to 20% of cases [277], most frequently in SOD1 ALS and PLS (50%) [278]. Urinary incontinence frequency ranges between 4% and 33%, especially in the form of urge incontinence, frequent in patients older than 60 years of age [279]. Aside from age, the predominance of upper motor neurons is also associated with urge incontinence, but whether this is directly related to the pathophysiology of the disease or the greater use of anticholinergic drugs or muscle relaxants for this particular phenotype is not clear. Erectile dysfunction in male ALS patients has also been reported [241].

Sialorrhea is a common feature affecting up to 20% of ALS patients, and aside from dysphagia, there is evidence that autonomic disturbances play a role in this disabling symptom. Although an excess of saliva in the mouth may impair salivary function, the primary secretory function of submandibular and parotid glands seems to be preserved. However, the inability to elicit a good response through indirect stimulation [280], and salivary gland function assessed with quantitative scintigraphy with 99mTc-pertechnetate [281], suggest alterations in the neuroendocrine mechanisms that regulate the secretory activity, even at early stages of the disease, and independently of the severity of motor symptoms. The lacrimal function has also been described as being decreased in ALS [282].

Regarding intestinal dysfunction, constipation has a high prevalence (about half) among ALS patients, but other factors beyond autonomic gastrointestinal dysfunction may play an important role, such as the lack of activity, inadequate content of fiber in the diet, and poor fluid intake (secondary to dysphagia). Nevertheless, functional explorations have found delayed gastric emptying [283,284] and delayed colonic transit time [285], as well as other disturbances in gastric and esophageal motility [89,286], all suggesting autonomic gastrointestinal involvement. In this regard, it is worth mentioning that these gastrointestinal abnormalities are found even in patients without dysphagia. In some patients with severe autonomic manifestations, beyond typical ALS neuropathological features, alpha-synuclein-positive intracytoplasmic inclusions have been found in ganglion cells of the esophageal nerve plexus [287], suggesting that in selected cases, alpha-synucleinopathy, and therefore an added early/presymptomatic parkinsonism, may be related, at least in part, with the autonomic dysfunction.

Although there is considerable evidence for the different impairments in the autonomic control of different target organs, some are characteristically spared, such as pupil responses [247,288]. In impaired organs, both sympathetic and parasympathetic systems are affected, resulting mostly in hypofunction that worsens over time. Sweat and cardiovascular features present increased activity during the early stages of the disease, probably reflecting the complexity of interactions during autonomic control dysfunction. Sympathetic over-activity could also be explained by the involvement of the limbic motor system and other central autonomic network structures (insular cortex, anterior cingulated gyrus, and basolateral nuclei of the amygdala) [289,290].

Pathological evidence points to preganglionic autonomic neurons as the main group of cells involved in autonomic dysfunction in ALS. Primary autonomic control centers, dorsal vagal, and solitary nuclei are seen to be spared in ALS [291,292]. Exceptionally in SOD1, V118L, and C146R-ALS cases, there is evidence of marked neuronal loss in dorsal vagal, solitary tract nuclei, and ambiguous nuclei [293,294], and in some cases, TDP-43 inclusions are present in the ambiguous nucleus which drives parasympathetic heart control [287]. Morphometrical assessment of the vagus nerve showed a preserved density of myelinated and unmyelinated fibers, supporting the hypothesis that parasympathetic autonomic disturbances are not explained by vagal deafferentation [295].

In the sympathetic system, the loss of neurons in the intermediate lateral column (IML) of the spinal cord has been reported in ALS patients [296,297], as well as alterations in the protein metabolism of this group of neurons, but with the preservation of the sympathetic ganglion cells [298]. IML neuronal loss typically begins in the upper spinal cord and progresses caudally toward lower thoracic segments [299]. No cytoplasmic aggregates or Bunina bodies have been described in IML neurons [300]. In SOD1^G93A^ mice, down-expression of choline acetyltransferase (ChAT) in superior cervical ganglia and tyrosine hydroxylase (TH) in the adrenal gland also imply preganglionic sympathetic denervation, which can cause trans-synaptic postganglionic dysfunction.

Histopathology studies have also shown a neuronal loss in Onuf’s nucleus in the ventral horn of the spinal cord, indicating an alteration in bowel and bladder control [301,302,303]. Only in sweat gland dysautonomia is there evidence of postganglionic impairment. This is of special interest, since postganglionic axons innervating eccrine sweat glands, although sympathetic, are also cholinergic, like all preganglionic autonomic neurons and somatic motor neurons.

Despite the abovementioned neuronal loss of autonomic structures in the central nervous system, the functional repercussion is mild in contrast to the motor manifestations. This is similar to other motor neuron disorders, such as Kennedy’s disease, where patients do not usually show autonomic symptoms, yet there is pathological evidence of subclinical impairment of parasympathetic and sympathetic neurons [304]. In contrast, in spinal muscular atrophy (SMA), histological studies did not find the involvement of the autonomic nervous system [305].

Disparities between functional and tissue involvement in ALS can be explained by reduced physical activity and muscular atrophy, but a different expression of glutamate receptor profiles between somatic and autonomic motor neurons could also be, in part, responsible. While ionotropic glutamate receptors may trigger excitotoxicity, group I metabotropic glutamate receptors (mGluRs1 and 5) exert neuroprotective effects. Parasympathetic Onuf’s nucleus and thoracic autonomic motor neurons express higher levels of these neuroprotective receptors (mGluR5), contrasting with their absence in somatic motor neurons [306].

## 16. Distal Axonopathy in ALS

A detailed look at the different sensory systems, both in patients and in animal models, demonstrates that ALS goes far beyond the motor system. Sensory neurons are more diverse in function, morphology, and molecular expression than motor neurons. For this reason, the study of selective vulnerability between different sensory neuronal populations may be of particular interest.

The resistance of the sensory neuron body is in line with the classical conception of the sparing of the main sensory pathways in ALS. Data from necropsies revealed that DRG neuronal loss is scarce and most likely affects large neurons [91]. Marked DRG neuronal loss has been reported in genetic forms in both human and animal models, also including pronounced sensory symptoms, predominantly in the form of gait ataxia [92,102,103]. In this regard, the DRG sensory neurons of SOD1^G93A^ mice are not decreased [99,100] but show structural alterations, seen under electron microscopies, such as swollen mitochondria, cytoplasmic fragmentation, and microvacuolization [95,96]. Sensory involvement is not an all-or-nothing phenomenon, but rather a graded phenomenon.

Different data have suggested the existence of sensory distal axonopathy in ALS. In the SOD1^G93A^ mouse, there is a gradient of loss of sensory axons from the epidermis through the dermis that increases over time [167]. This loss of IENF is not followed by a loss of their respective neuronal bodies in the DRG [100]. In line with this, a decreased sudomotor innervation in the skin of both ALS patients and animal models has been described [100,161,165,257], but sympathetic sudomotor ganglion cells are preserved [298]. Similarly, the distal portion of proprioceptive sensory fibers in the muscle spindles degenerate in SOD1^G93A^- and TDP43^A315T^-mutated mice already at a presymptomatic stage [136], and in the drosophila SOD1^G85R^ model [138], before motor neuron degeneration is established, while proprioceptive neurons in the DRG are spared [99,100].

A distal axonopathy feature has been described in the motor counterpart of ALS, in animal models and human patients, with the early loss of neuromuscular synapses occurring even before a-motor neurons in the anterior horn of the spinal cord are lost [101,307,308,309,310,311,312]. As an example, in SOD1^G93A^ mice, denervation of the tibialis anterior muscle occurs 14 to 30 days prior to evidence of motor neuron degeneration [313,314]. In the *FUS* mice and *FUS* drosophila models, there are also early synaptic changes at the neuromuscular junction that lead to progressive denervation [308,309]. Other findings that support axonal contribution early in the disease include the presence of morphologic abnormalities in distal motor axons, varicosities along the intramuscular nerve fibers, terminal swellings, and even the presence of sprouting in the absence of significant denervation in SOD1 *knockout* mice [310].

Although the loss of small fibers in the skin is similar in the distal leg and thigh in ALS patients [161], this does not argue against the distal axonopathy model. Length-dependent axonal disturbances translate into a gradient of manifestations from distal (more profound) to proximal (less affected), but certain variability may occur. In line with this, SOD1 knockout mice showed that motor axons with similar length degenerate differently [310]; tibialis anterior muscle became significantly denervated (30%) by four months, while in soleus muscle, no significant denervation was seen until 12 months of age. In this case, axon length was not a determining factor, and muscle type/motor unit was more relevant, with the fast-twitch muscles of the hind limbs exhibiting greater atrophy over time than slow-twitch muscles [313,314].

The role of early denervation of the neuromuscular junction has raised the possibility of non-autonomous cell death as a key factor in the neurodegeneration process in ALS. As an example, the muscle expression of Nogo-A is correlated negatively with disease duration, and positively with disease progression rate and proportion of denervated neuromuscular junctions [221,312,315]. Another link has postulated that the expression of semaphorin 3A (Sema3A), an axonal chemorepellent molecule, in the terminal Schwann cells associated with the neuromuscular junction of the most vulnerable fast-fatigable motor units explains initial denervation and later motoneuron death [316].

Beyond the possible role of skeletal muscle and terminal Schwann cells in the denervation process, different mechanisms have been described as contributing to ALS distal axonopathy. Deficits in axonal transport are reported to contribute to many neurodegenerative diseases. Axonal function depends heavily on a healthy cytoskeleton composed of microfilaments and intermediate filaments (predominantly for structural support) and microtubules. Aggregates or abnormalities in neurofilaments have been directly linked to the ALS phenotype, and pharmacological stabilization of microtubules decreases motor neuron death and improves life expectancy in ALS mice [317]. In ALS mouse models, deficits in both retrograde and anterograde transport have been reported, even independently of denervation [313,314,318,319]. In SOD1^G93A^ mice, there is a significant impairment in retrograde transport at a presymptomatic stage, which worsens over time [320]. In vivo analysis showed the altered transport of mitochondria in the presymptomatic SOD1^G93A^ mouse [318]. However, no early differences in the rate of retrograde transport in either tibialis anterior or soleus innervating axons in SOD1 mice were found.

Some genetic forms of ALS have links with axonal function. Genes related to monogenic forms of the disease can be classified into four groups depending on the cellular pathways that are involved: protein homeostasis, RNA homeostasis, and trafficking, mitochondrial function, and cytoskeletal dynamics [321,322,323,324,325,326].

In addition to deficits in axonal transport, other mechanisms can explain pathological findings such as axonal swellings found in sensory axons. In physiological conditions, axonal local translation contributes to axonal efficiency and compartmentalization [327]. Whereas terminal swellings seen in ALS are also seen in other models of motor neuropathy and Wallerian degeneration, they are usually related to protein aggregation and post-translational alterations and are not due only to transport deficits. In this regard, it has recently been revealed that TDP-43 controls the axonal transport of ribosomal protein mRNAs and local translation by ribosomes, essential to maintain the morphological integrity of axons [328].

In addition, it is hypothesized that the distal portion of axons could have an intrinsic susceptibility to oxidative stress, and/or that microenvironment conditions surrounding distal axons could present increased oxidative stress [329,330,331]. Interestingly, the SOD1 *knockout* mouse, a model of chronic oxidative stress, was originally thought to lack a motor phenotype until it was discovered that these mice depict a model of distal axonopathy, with the pathology restricted to the distal axon, and very early involvement of the fast-twitch muscles [310]. In this animal, epidermal sensory fibers are characteristically preserved, in contrast with the SOD1^G93A^ mice [167]. This suggests that sensory axons could be more resilient or that they are less exposed to oxidative stress.

Gain-of-function *SOD1* mutation could contribute to sensory axonopathy [332,333,334,335,336,337]. Toxic effects in the distal portion of axons of mutated *SOD1* have been suggested based on the improvement and increased survival in SOD1^G93A^ and Loa-crossbred mice (an animal model with mutated cytoplasmic dynein that results in deficits in the retrograde transport) [338,339], although similar results were not found in SOD1^G85R^ and SOD1^G73R^ transgenics [147].

## 17. Conclusions

Although generally irrelevant from a clinical point of view, sensory involvement is widely present in ALS. The selectivity of motor vulnerability has been the cornerstone of research on the disease and we now understand that such vulnerability does not comprise a qualitative but rather a quantitative characteristic. Sensory involvement may not only contribute to the understanding of the disease as a systemic one, but it also opens new possibilities for the study of the pathophysiological mechanisms of the neurodegenerative process. There is a general involvement of somatic small and large sensory axons, and of autonomic axons in ALS, which progresses over time. A parallel study of their neuronal bodies shows a pattern of distal sensory axonopathy, with degeneration of the most distal portion of the axon while the soma is preserved. A similar pattern of degeneration has been accepted in the neuromuscular system in ALS, reflecting similarities in terms of disease pathophysiology between such diverse neurons.

Sensory pathways should be explored more often and investigated in more detail in research in both human and animal models. For instance, the presence of TPD-43 and its phosphorylated form and post-translational modifications should be studied further in sensory axons and their respective bodies, to improve the understanding of the differences and similarities with the motor system. Sensory axons are also more accessible than motor fibers, so their role as a tool for diagnosis or even for drug response monitoring should be examined. In recent years, there has been an increase in the number of genes shown to be involved in ALS as well as in other neurodegenerative diseases with different phenotypes that include clinically relevant sensory involvement. This represents a valuable opportunity to study new shared disease mechanisms, and hopefully, this will contribute to new avenues for the treatment of this devastating disease.

## Figures and Tables

**Figure 1 ijms-23-15521-f001:**
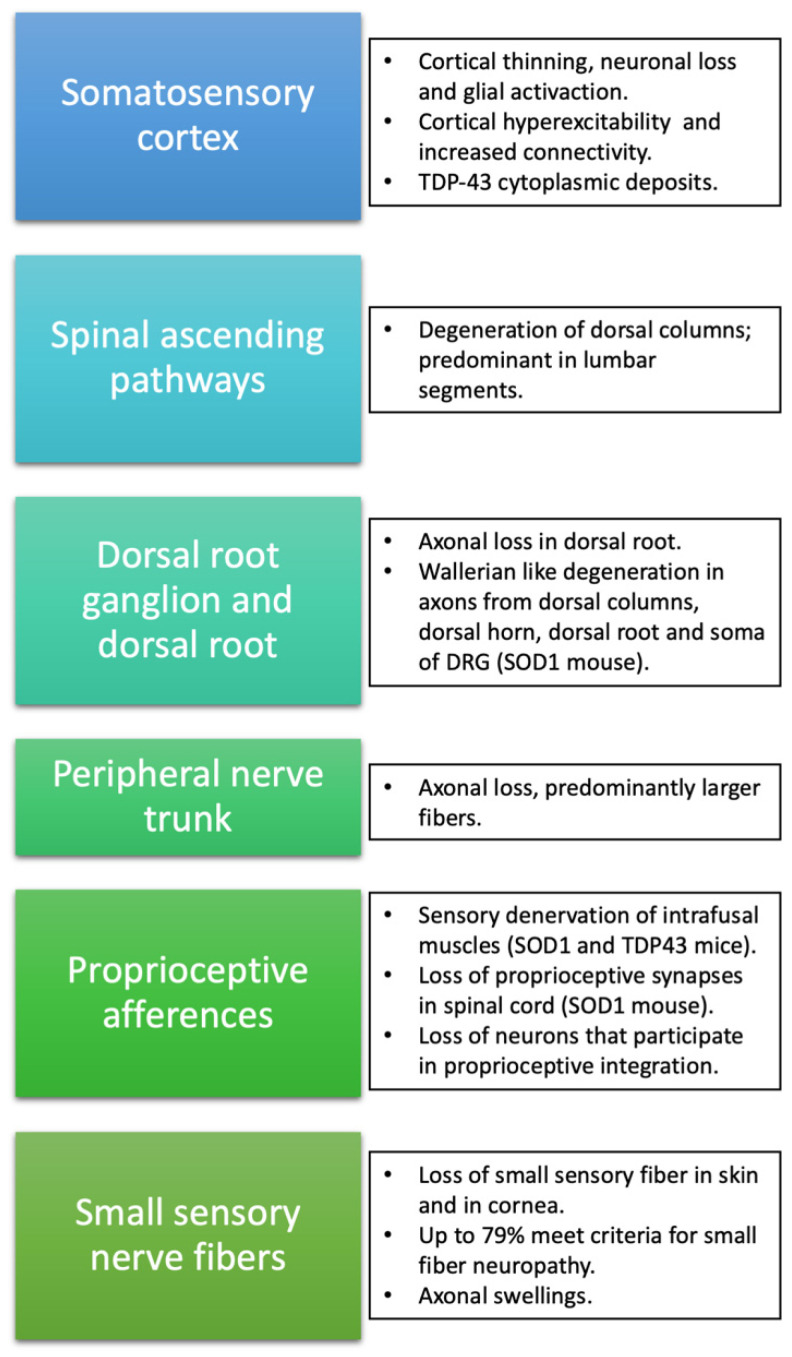
Schematic summary of alterations described in ALS at the different levels of the somatosensory system.

**Figure 2 ijms-23-15521-f002:**
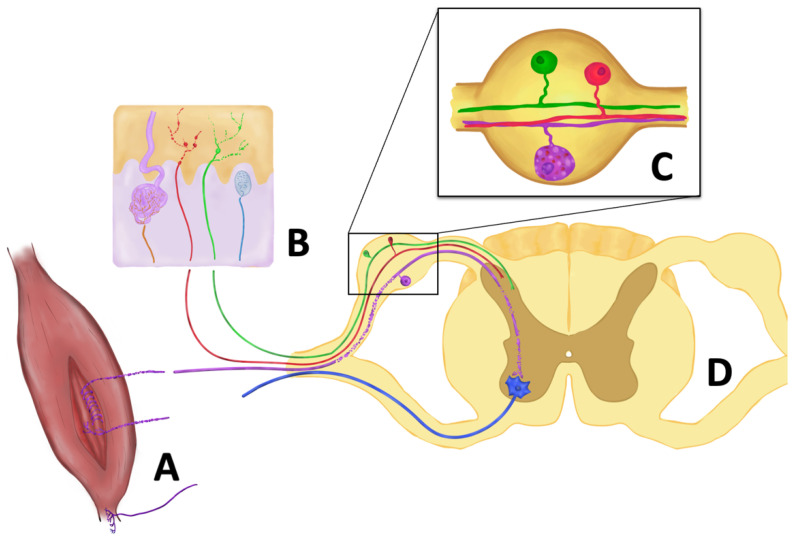
Main pathological changes in sensory afferents reported in ALS. (**A**) Degeneration of type Ia and II sensory fibers innervating muscle spindles, while type Ib fibers (Golgi tendon organ) are preserved. (**B**). Reduction of intraepidermal nerve fibers, terminals of both peptidergic (red) and non-peptidergic (green) axons, with focal swellings. Meissner corpuscle innervation (blue) is also impaired, as well as sympathetic fibers innervating sweat glands (brown). (**C**) In the dorsal root ganglion, neuronal bodies are preserved, although vacuolization and accumulation of misfolded SOD1 protein are seen in proprioceptive neurons (purple). (**D**) In the dorsal root, larger axons show Wallerian degeneration. In addition, proprioceptive synapses in the anterior horn are reduced in certain animal models (SOD1^G93A^ mouse) but preserved in others (TDP43^A315T^ mouse).

**Figure 3 ijms-23-15521-f003:**
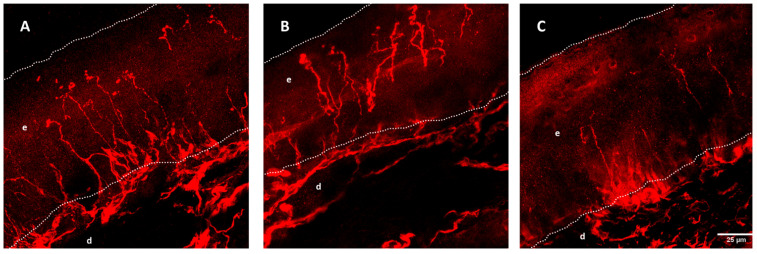
Intraepidermal nerve fiber (IENF) density from wild-type (**A**), SOD1^G93A^ mice of 8 (**B**), and 16 weeks (**C**). Dotted lines mark the limits of the epidermis (e) and dermis (d). IENF fibers are marked with PGP9.5 and show a decrease over time in the SOD1^G93A^ mouse. Reprinted/adapted with permission from [167], 2016, John Wiley and Sons.

## Data Availability

Not applicable.

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
