# Peer review of "Sensory Involvement in Amyotrophic Lateral Sclerosis"

_ijms, 2022, doi:10.3390/ijms232415521_

Round 1

Reviewer 1 Report

The manuscript (ijms-1996049) entitled “Sensory involvement in amyotrophic lateral sclerosis,” by Rubio, Herrando-Grabulosa, and Navarro review the of non-motor manifestations, including sensory and autonomic systems, and cognition, in patients with amyotrophic lateral sclerosis (ALS). They also review the involvement of other system in ALS including somatosensory, visual, olfactory, gustatory, and auditory. This could be a very interesting and needed publication, but in its current form falls short of this expectation.

The authors are correct in stating that “ALS is eminently a motor disease,” but they lose my enthusiasm when they write “sensory symptoms are not of great relevance for ALS patients.” They even conclude with the statement that “although generally irrelevant from a clinical point of view …” This is incorrect, at least it is in our clinic. For clinicians and researchers in the ALS community, these non-motor manifestations are very relevant. It is now widely accepted that ALS is a systemic disorder. The authors write that “the most clinically relevant non-motor feature is cognitive impairment, present in approximately 30-50% of patients.” This is correct and needs emphasized and expanded upon. This reviewer thinks that the authors rely too heavily upon animal data and should concentrate primarily upon patient derived data.

            The manuscript is difficult to read. There are many sentences that do not make sense. For instance, line 39 and forward, the authors write “In these, it is discussed the role of sensory neurography to rule out other pathologies, pointing out that for an ALS diagnosis,…” Line 52 “It is unclear whether these extra-motor abnormalities occur in all patients, such that ALS can be considered a multisystem disorder, although with a core set of motor symptoms required for diagnosis [7,8].” Such grammatical errors occur throughout the manuscript.

            The authors write that “ALS patients are usually immobilized and confined to bed or a wheelchair, but they rarely develop bedsores, even in terminal stages.” The beginning clause of the sentence is incorrect. Patients are not in wheelchairs or bedridden until later in disease. The clause of this sentence is correct. On line 153, it should read “other investigators” and not “other authors.” On line 160, it is not “On this regard,” but “in this regard.” By convention, all paragraphs should be at least three sentences. The authors need an editor fluent in English and English grammar for this manuscript to be publishable.

Author Response

Dear editor,

We thank the reviewers and editor for their comments, as well as for the time dedicated to make constructive suggestions to improve the manuscript.

Based on the reviewers’ feedback we have made some changes that we believe that have introduced significant improvements in several parts of the text.

Below you will find the detailed list of changes to make as clear as possible the changes that we included into this revised version of the manuscript.

Figure 3 has been changed. This replacement is according to the standard of the ‘Review’ type article of IJMS, as agreed upon with the assistant editor.

Response to Reviewer 1:

Changes:

  • Reviewer comment: The authors are correct in stating that “ALS is eminently a motor disease,” but they lose my enthusiasm when they write “sensory symptoms are not of great relevance for ALS patients.” They even conclude with the statement that “although generally irrelevant from a clinical point of view …” This is incorrect, at least it is in our clinic. For clinicians and researchers in the ALS community, these non-motor manifestations are very relevant.
  • Response/change: We agree with the Reviewer. Indeed, we wrote this paper to provide the clinicians and researchers interested in MND, a comprehensive review emphasizing the presence and relevance of sensory and other non-motor symptoms and signs in ALS. Thus, we have changed the Abstract so not undervalue the clinical significance of sensory and non-motor manifestations in ALS. The sentence now reads:
    • ‘Although from a clinical perspective, sensory symptoms are overshadowed by the motor manifestations, this does not mean that their pathological significance is not relevant.’

  • Reviewer comment: The authors write that “the most clinically relevant non-motor feature is cognitive impairment, present in approximately 30-50% of patients.” This is correct and needs emphasized and expanded upon.
  • Response/change: We have added more clinical data about cognitive impairment in ALS as well as its references.

“The cognitive profile of these patients is described by deficits in executive functions (especially phonemic verbal fluency that occurs early), language, social cognition, and verbal memory [16]. Language impairment could be minimized due to confound factors such as dysarthria and respiratory insufficiency. Some characteristic features are syntactic processing deficits, semantic, verbal, and grammatical errors [24-29]. Behavioral changes are also described, being apathy the most common feature (40-80%). Depression around the time of diagnosis is also common, and other traits can appear such as disinhibition, irritability, loss of empathy, and emotional lability [30-32].

Frontotemporal lobar degeneration explains cognitive impairment only in a part of ALS patients, while other features such as the extent of TDP-43 pathology and the presence of hippocampal sclerosis may also contribute to cognitive disturbances [33].”

  • Reviewer comment: This reviewer thinks that the authors rely too heavily upon animal data and should concentrate primarily upon patient derived data.
  • Response/change: More clinical information has been added in the extra-motor section as mentioned previously. However, the aim of the review is to bring together information derived from patients as well as molecular and/or animal model data to have a holistic view of what really happens in the disease. Although it is certain that there are some specific sections where there is much more information on animal models (dorsal root ganglion and proprioceptive afferences), this is because the information from patients with affectations at these levels is scarce, and the experimental models are very valuable to investigate alterations and mechanisms.

  • Reviewer comment: There are many sentences that do not make sense. For instance, line 39 and forward, the authors write “In these, it is discussed the role of sensory neurography to rule out other pathologies, pointing out that for an ALS diagnosis,…” Line 52 “It is unclear whether these extra-motor abnormalities occur in all patients, such that ALS can be considered a multisystem disorder, although with a core set of motor symptoms required for diagnosis [7,8].”
  • Response/change: Those suggested changes has been made.
    • “According to those criteria sensory neurography should be normal (in both amplitude and velocity) unless an entrapment neuropathy or another known etiology could explain its abnormalities. “
    • “Although it is unclear whether these extra-motor abnormalities occur in all patients, ALS can be considered a multisystem disorder, with a core set of motor symptoms required for diagnosis [7,8].”

  • Reviewer comment: The authors write that “ALS patients are usually immobilized and confined to bed or a wheelchair, but they rarely develop bedsores, even in terminal stages.” The beginning clause of the sentence is incorrect. Patients are not in wheelchairs or bedridden until later in disease. The clause of this sentence is correct.
  • Response/change: Thanks for the comment. We have pointed out that this situation occurs when patients are in an advanced stage of the disease.
    • ‘ALS patients in later stages of the disease are usually immobilized and confined to bed or a wheelchair, but they rarely develop bedsores, even in terminal stages’.

  • Reviewer comment: On line 153, it should read “other investigators” and not “other authors.” On line 160, it is not “On this regard,” but “in this regard.”
  • Response/change: Those changes have been made in the revised version of the manuscript.

  • Reviewer comment: By convention, all paragraphs should be at least three sentences.
  • Response/change: Paragraph extension has been revised to avoid too long or too short sections.

  • Reviewer comment: The authors need an editor fluent in English and English grammar for this manuscript to be publishable.
  • Response/change: Manuscript has been revised and improved by a native English corrector.

Reviewer 2 Report

This is an excellent and comprehensive review of sensory and other non-motor involvement in ALS. It draws on all multidisciplinary modalities of investigation, as well as human ALS syndromes and animal models. The authors are presumably not native English-speakers, and there are a number of instances of phrases that are awkward in the English translation. Some of these are pointed out in the comments to the Editors.

The manuscript needs careful editing by an English-speaking editor to eliminate a number of awkward phrases and sentences that have resulted from translation from the original Spanish. Examples are:

p256: "sensitive" - should be sensory.

p287: "mutation heterozygosity" - should be heterozygous mutations.

p390: "no" should be not.

p722: "large evidence" - should be considerable evidence.

728: "sympathetic over function - should be over-activity.

p782-784: replace sentence with: 'Sensory involvement is not an all-or-none phenomenon, but rather a graded phenomenon."

For the sake of giving reference to earlier literature, the authors should be encouraged to include reference to: Bradley et al. 1983 Morphometric and biochemical studies of the peripheral nerves in ALS. Neurology 1983: 14; 267.

Author Response

Dear editor,

We thank the reviewers and editor for their comments, as well as for the time dedicated to make constructive suggestions to improve the manuscript.

Based on the reviewers’ feedback we have made some changes that we believe that have introduced significant improvements in several parts of the text.

Below you will find the detailed list of changes to make as clear as possible the changes that we included into this revised version of the manuscript.

Figure 3 has been changed. This replacement is according to the standard of the ‘Review’ type article of IJMS, as agreed upon with the assistant editor.

Response to reviewer 2:

  • Reviewer comment: The manuscript needs careful editing by an English-speaking editor to eliminate a number of awkward phrases and sentences that have resulted from translation from the original Spanish.
  • Response/change: Manuscript has been revised and improved by a native English corrector.

  • Reviewer comment: p256: "sensitive" - should be sensory. / p287: "mutation heterozygosity" - should be heterozygous mutations. / p390: "no" should be not. / p722: "large evidence" - should be considerable evidence. / 728: "sympathetic over function - should be over-activity. / p782-784: replace sentence with: 'Sensory involvement is not an all-or-none phenomenon, but rather a graded phenomenon."
  • Response/change: Those changes has been introduced in the revised version of the manuscript. We appreciate your help.

  • Reviewer comment: For the sake of giving reference to earlier literature, the authors should be encouraged to include reference to: Bradley et al. 1983 Morphometric and biochemical studies of the peripheral nerves in ALS. Neurology 1983: 14; 267.
  • Response/change: Thanks for the suggestion. We have added the reference and more information of this paper in the manuscript.
    • “Moreover, there is evidence of a reduction of axonal transport in sensory axons of sural nerves [110].”

Round 2

Reviewer 1 Report

The authors have satisfactorily answered my concerns.